# CoPheScan: phenome-wide association studies accounting for linkage disequilibrium

Ichcha Manipur [1,2] ✉, Guillermo Reales [1,2], Jae Hoon Sul[3], Myung Kyun Shin [3], Simonne Longerich[3], Adrian Cortes[4] & Chris Wallace [1,2,5]

Phenome-wide association studies (PheWAS) facilitate the discovery of associations between a single genetic variant with multiple phenotypes. For variants which impact a specific protein, this can help identify additional therapeutic indications or on-target side effects of intervening on that protein. However, PheWAS is restricted by an inability to distinguish confounding due to linkage disequilibrium (LD) from true pleiotropy. Here we describe CoPheScan (Coloc adapted Phenome-wide Scan), a Bayesian approach that enables an intuitive and systematic exploration of causal associations while simultaneously addressing LD confounding. We demonstrate its performance through simulation, showing considerably better control of false positive rates than a conventional approach not accounting for LD. We used CoPheScan to perform PheWAS of protein-truncating variants and fine-mapped variants from disease and pQTL studies, in 2275 disease phenotypes from the UK Biobank. Our results identify the complexity of known pleiotropic genes such as *APOE*, and suggest a new causal role for *TGM3* in skin cancer.

Phenome-wide association studies (PheWAS) are an inversion of the GWAS (Genome-Wide Association Studies) paradigm, where a single genetic variant is tested against a broad range of phenotypes. Phenome scale studies are facilitated by the availability of a broad array of phenotypes linked to genomic data in large-scale biobanks. PheWAS are a promising tool in the field of pharmacogenomics as they facilitate drug repurposing efforts and identification of potential adverse effects due to their ability to detect pleiotropy[1–3]. Often, PheWAS has been paired with other approaches such as Mendelian randomisation to identify causal effects of exposures on outcomes and network analysis to identify interactions between phenotypes[4–6].

Prevailing methods for phenome-wide testing are built upon single variant tests and do not inherently tackle the spurious associations that can arise when traits are causally associated not with the index variant, but with another variant in LD with the index variant. For instance, a PheWAS of UK Biobank phenotypes with protein-truncating variants by DeBoever et al.[7] first revealed an association between an *ANKDD1B* variant, and high cholesterol, which was found to reflect an indirect association, through LD with an intronic variant in *HMGCR* which is known to be associated with cholesterol levels. Thus, LD confounding necessitates the use of additional follow-up tests such as colocalisation analyses, where pairs of traits are tested for shared causal variants within a genomic region, to isolate associations that are truly causal[3,8].

PheWAS hits are colocalised with molecular QTLs or disease traits on which the identified variants have a prior known effect. However, this two-step approach is not feasible for variants with known biological effects for which summary statistics are unavailable, such as those involved in protein truncation.

[1]Cambridge Institute of Therapeutic Immunology & Infectious Disease (CITIID), Jeffrey Cheah Biomedical Centre, Cambridge Biomedical Campus, University of Cambridge, Cambridge CB2 0AW, UK. [2]Department of Medicine, University of Cambridge School of Clinical Medicine, Cambridge Biomedical Campus, Cambridge CB2 2QQ, UK. [3]Merck & Co., Inc., Rahway, NJ, USA. [4]Human Genetics and Genomics, GSK, Heidelberg 69117, Germany. [5]MRC Biostatistics Unit, University of Cambridge, Cambridge, UK. ✉e-mail: im504@cam.ac.uk

In this work, we introduce a Bayesian approach to PheWAS, Coloc adapted Phenome-wide Scan, (CoPheScan), that tests phenome-scale causal associations with a set of index variants while handling confounding due to LD at the same time. CoPheScan can exploit external covariate data, such as the genetic correlation between phenotypes, and can be run in different ways depending on whether accurate LD information is available and whether the analyst is prepared to make assumptions about the number of causal variants in the tested genomic region. We demonstrate the utility and robustness of these different approaches on simulated datasets. We also analysed causal variants selected from three real-world sources and tested for causal associations against 2275 phenotypes from the UK Biobank using CoPheScan.

## Results

### Overview of CoPheScan

CoPheScan is an adaptation of the coloc[9–11] approach, for the case where a variant known to be causal either through fine-mapping or functional studies, is subjected to a phenome-wide scan to test for causal associations with other phenotypes/traits. Coloc considers genetic association patterns for two traits in a genomic region and assesses whether it is likely they share a causal variant in that region. It is a Bayesian approach and assumes prior probabilities for each of the five possible hypotheses (no association with either trait, association with just one trait or the other, association with both traits and different causal SNPs, or association with both traits at the same causal SNP) are fixed and known.

We consider the case where a SNP of interest is known to be causal for a phenotype which is often the case in PheWAS, and we are interested in determining if it is also causally associated with another phenotype (Fig. 1a). We will hereafter refer to the variant of interest as the query variant, the phenotype for which the query variant is known to be causally associated as the primary trait and the phenotype to be tested as the query trait. In a genomic region with $Q$ SNPs, and under the initial assumption of a single causal variant (which we will relax later), there are $Q+1$ possible ways or "configurations", (Supplementary Fig. 1) to describe where the single causal variant may lie, each corresponding to exactly one of three hypotheses:

$H_n$: No association of any variant with the query trait (one configuration)

$H_a$: Causal association of a variant other than the query variant with the query trait ($Q - 1$ configurations)

$H_c$: Causal association of the query variant with the query trait (one configuration)

The posterior odds for each hypothesis ($H$) given the data ($D$) for the query trait with respect to the null hypothesis ($H_n$) is given by,

$$\frac{P(H|D)}{P(H_n|D)} = \frac{P(H)}{P(H_n)} \times \frac{P(D|H)}{P(D|H_n)} \tag{1}$$

In Eq. (1), the first ratio in the right-hand side is the prior odds and the second ratio is the Bayes Factor (BF). Thus, the three prior probabilities that have to be specified are: $p_n = p(H_n)$, $p_a = p(H_a)/(Q-1)$, and $p_c = p(H_c)$, subject to the constraint that $p_n + (Q-1)p_a + p_c = 1$.

Beyond the difference in the hypothesis space described above, CoPheScan differs from coloc in two further ways. First, because we have reduced the hypothesis space, we can examine many variants simultaneously, allowing us to learn the priors from the data in a hierarchical Bayesian manner with Markov Chain Monte Carlo (MCMC) sampling (Supplementary methods). In contrast, coloc assumes priors are fixed and known, which is a weakness because inference must rely on the investigators' judgement on prior probabilities of colocalisation. Second, because we are using this hierarchical approach, we can exploit additional external information about the variants and/or the traits in the form of covariates which can be included when learning

the priors. This allows the priors to vary depending on the query trait/query variant pairs being considered. Here, we include the genetic correlation ($r_g$) between the primary trait and each query trait tested (see Supplementary Methods).

The restriction to a single causal variant allows us to count the possible configurations ($Q+1$), and if the assumption is deemed valid, CoPheScan can be run directly on summary GWAS data using Wakefield's method[12], to compute approximate Bayes factors summarising the relative support for a model where the SNP is associated with a trait compared to the null model of no association. However, this assumption is not broadly valid, and an alternative is to use the Sum of Single Effects (SuSiE) Bayesian fine mapping regression framework[13,14] to partition the evidence into configurations corresponding to each of multiple possible causal variants and use these in a similar manner to allowing for multiple causal variants in coloc[10]. The SuSiE approach works best with either raw genotype data or summary GWAS data when in-sample LD information is available[15].

Hence, CoPheScan has the flexibility to be run in several ways (Fig. 1b) depending on: (i) the assumption about the number of causal variants, (ii) the specification of either fixed or hierarchical priors, and (iii) the inclusion/exclusion of covariates if the hierarchical model is used to infer priors. A detailed description of the CoPheScan method is available in the Supplementary methods. A summary of the simulated data, variant and phenotype sources used for the analysis with the real data can be found in (Fig. 1c), while a detailed description is provided in the Methods.

### Simulations show CoPheScan is more accurate than a standard method which does not account for LD confounding

We simulated regional GWAS summary data for traits with either zero, one or two causal variants (Methods) such that they corresponded to the three CoPheScan hypotheses. We also allowed the probability of Hc to vary according to a simulated genetic covariance between primary and query traits and considered whether including this information in the analysis increased inferential accuracy. We analysed the same data in parallel using a conventional PheWAS approach of testing each of the set of query SNPs for association, controlling either the FDR or the family-wise error rate via Bonferroni correction. We compared these to the results from CoPheScan, using a hierarchical model (with and without the covariate data) or fixed priors chosen as described in the Supplementary methods which broadly matched the proportion of Hn, Ha, and Hc in the sample.

First, we considered the appropriate threshold on the posterior probability of Hc, ppHc, to call an association. We estimated the FDR internally, as 1-mean(ppHc) | ppHc > t for different values of threshold t (Supplementary Fig. 4). We found that ppHc > 0.6 maintained an FDR < 0.05 across all analyses of simulated data. Using this threshold, CoPheScan appeared less sensitive to the presence of a single causal variant (true Hc) than the conventional BH approach but more sensitive than the Bonferroni approach (Fig. 2). CoPheScan demonstrated control of the FDR (0.026–0.039) estimated as the proportion of significant calls that were truly Hn or Ha, traits where the query variant was not causal, for the different CoPheScan approaches compared to 0.219 and 0.308 for the conventional BH and Bonferroni approaches respectively, (Supplementary Data 1). The majority of the false positives obtained from these conventional approaches were true Ha but called as associated due to LD confounding. All CoPheScan approaches performed well in the case of a single causal variant, but when there were two causal variants (True Hc2), using SuSIE resulted in ~30% higher sensitivity to correct Hc predictions than the ABF approach (Fig. 2). This was balanced against marginally lower (<0.5%) sensitivity to Hc with SuSiE when traits truly had only a single causal variant (True Hc) when compared to the CoPheScan approaches that assumed a single causal variant.

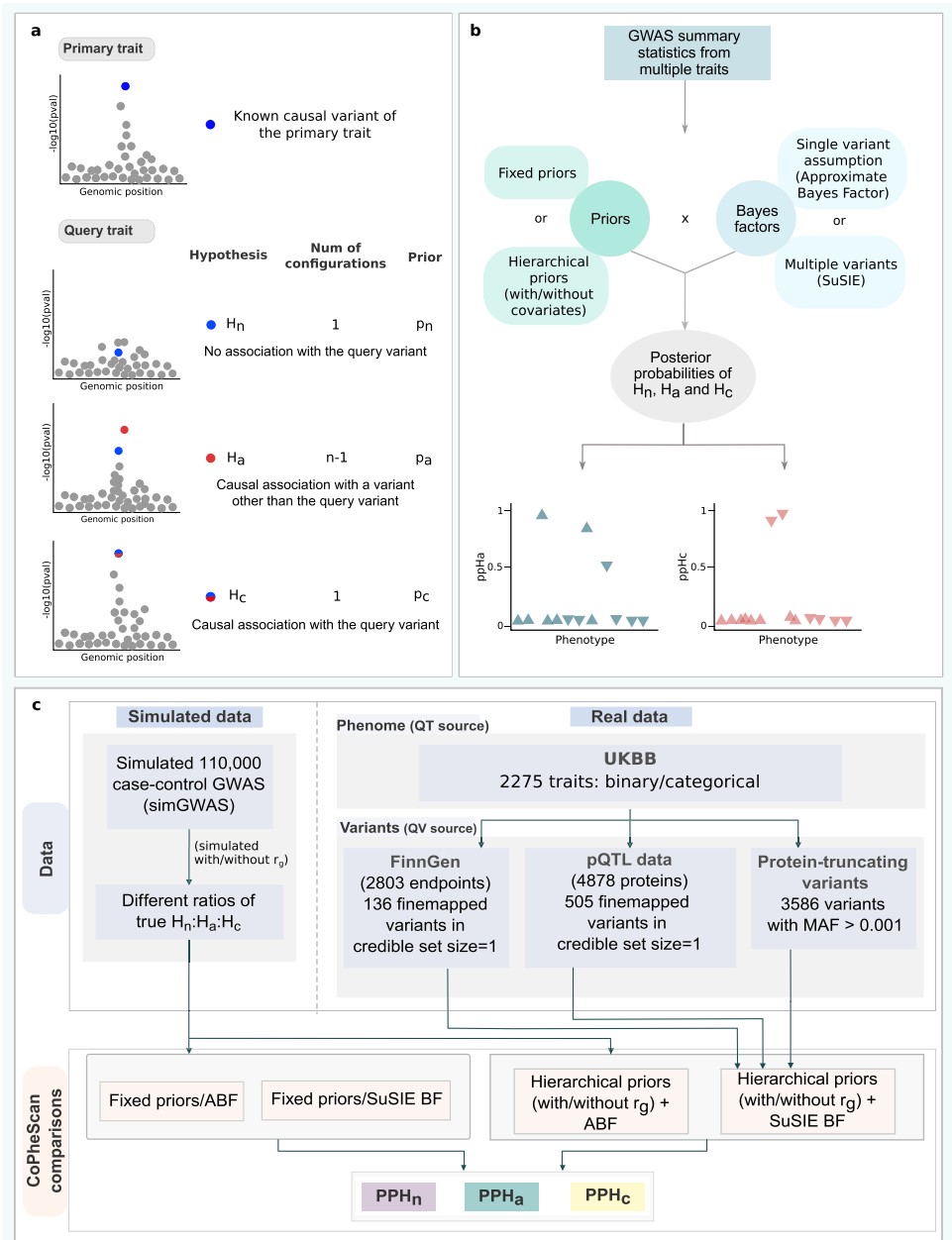

**Fig. 1 | Introduction and evaluation of the CoPheScan method. a** CoPheScan methodology: Hypotheses with illustrations of the configurations of genetic variants within the genomic region and corresponding priors. **b** Schematic of the CoPheScan workflow. The inputs are GWAS summary statistics from multiple traits and the position of the query variant. Computation of the posterior probabilities of the three hypotheses is performed with priors and Bayes factors computed using different CoPheScan approaches. **c** Study design for evaluation: Simulated data − Generated using SimGWAS and all CoPheScan approaches were run on this set. Real data − Phenotypes tested were obtained from UK Biobank and variants from fine-mapping FinnGen and a proteome dataset[25]. Hierarchical priors and SuSIE BF were used on the real data to identify SNP-disease associations. (QV - query variant, QT - query trait).

Although the effect of including covariate information was minor overall, Fig. 3a shows that it had a substantial effect in a minority of cases, bringing ppHc from below to above 0.6 in 2.79% of true Hc and Hc2 cases (80/2867), although also in 0.088% of true Hn and 0.011% of true Ha and Ha2.

Finally, these initial simulations showed that the hierarchical model recovered very similar results to the fixed prior model, where we chose our fixed prior values to broadly match the simulation scenarios, i.e., an optimal scenario. This offers reassurance that the hierarchical model can perform just as well as a method that "knows" the correct prior values. However, in real data, we will not know the true proportion of Hn, Hc, or Ha in our data, so we explored the robustness of both approaches to variations in these proportions. We found that using over-optimistic fixed priors, i.e. when the prior probability for Hc (P(Hc) = 0.091) exceeded the proportion of Hc in our data, led to dramatically high FDR, whilst the hierarchical model correctly adapted to the different datasets so that the FDR was controlled except at the very lowest true proportions of Hc (Fig. 3b).

## Using genetic correlation as a covariate increases detection of associations with disease-causal variants

We explored the performance of CoPheScan (Supplementary Fig. 5) using a variety of causal variants sets to perform PheWAS in three sets of query variants in up to 2275 query traits (Supplementary

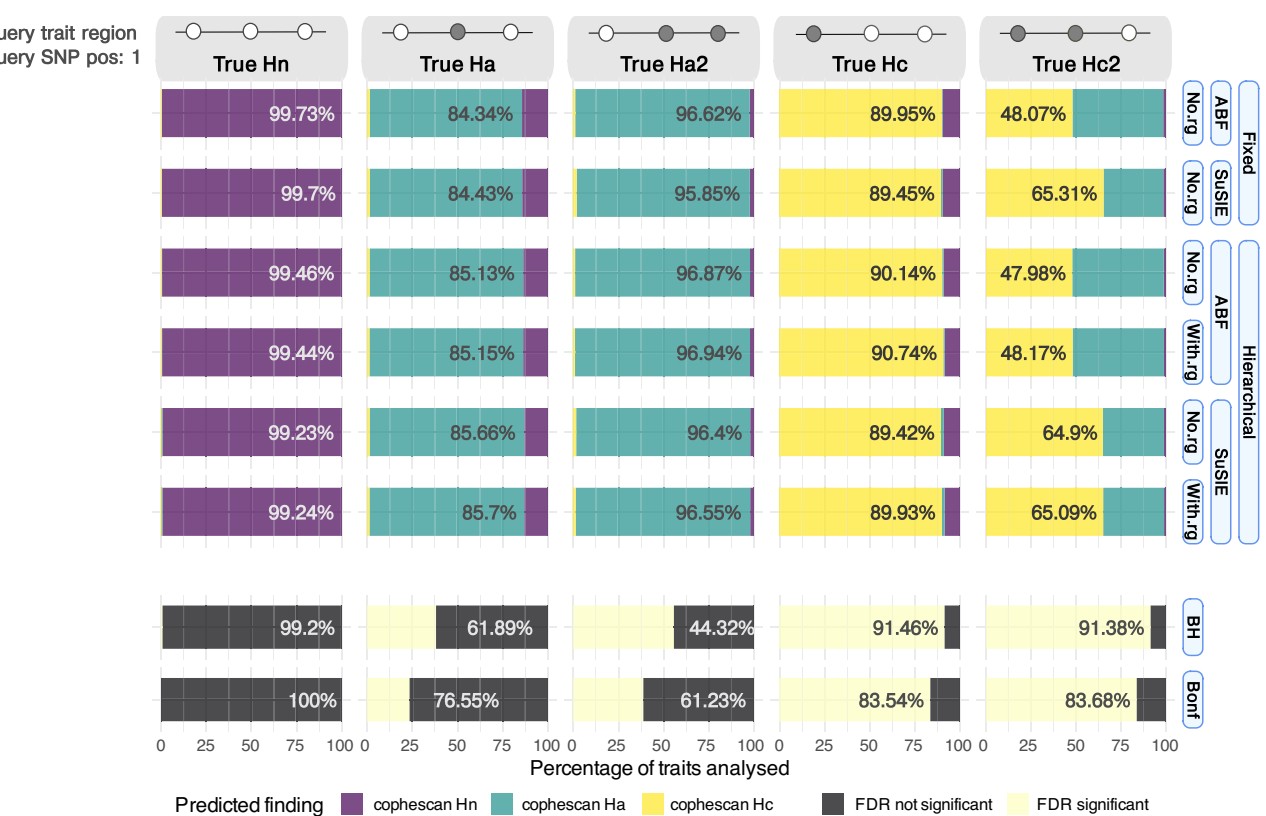

**Fig. 2 | Results for hypotheses discrimination in simulated data.** We called a single result for each simulated trait as described in Methods. The x-axis shows the percentage of hypothesis calls using the different approaches shown on the y-axis. For CoPheScan (top 6 rows), the three labelled columns on the y-axis, from right to left, indicate the type of priors used, the method used to calculate Bayes factors, and whether or not genetic correlation ($r_g$) was used. The last two rows show conventional approaches controlling the FDR (BH Benjamini-Hochberg) or the FWER (Bonf Bonferroni) at 0.05. The top bar shows an illustration of the configuration of SNPs in the genomic region corresponding to the different simulated traits (Methods), with the queried variant at position 1 and causally associated (non-associated) variants indicated by filled (open) circles. [True Hn: no causal variant, True Ha/Ha2: one/two causal non-query variants, True Hc: causal query variant, True Hc2: causal query variant and one causal non-query variant].

Figs. 6 and 7) from the UK Biobank summary data provided by the Neale Lab (http://www.nealelab.is/uk-biobank/). First, 136 disease-causal variants were identified as single variant credible sets in fine mapping data from FinnGen disease endpoints (primary traits, https://www.finngen.fi/en/access_results). We identified causal associations in UKBB at 43 (31.62%) of these, predominantly amongst query traits identical or related to the primary trait. Out of 101 unique query-variant-primary trait pairs with exact query variant-query trait matched pairs in UKBB, 32 were found to be Hc (Supplementary Fig. 8), and 65 Hn due to a lack of power in UKBB (p-value > $10^{-5}$). Four cases were called Ha, and in these the UKBB p-value was small, but the fine mapping produced different results in UKBB and FinnGen (Supplementary Fig. 8).

Genetic correlation ($r_g$) information for only 1582 out of the 2275 traits used in analysis without $r_g$ was available. $r_g$ values between the 1582 query traits and 69 UKBB traits which were matched with the FinnGen primary traits were used as a covariate (130697 query trait-query variant pairs tested). Including $r_g$ in the hierarchical model made a larger difference here than in the simulated data, perhaps reflecting a stronger effect than we anticipated in our simulations. Overall, ppHc values for traits with higher $r_g$ with the primary traits increased and, conversely, decreased for traits with lower (negative) $r_g$ (Fig. 4). Incorporating the $r_g$ resulted in the identification of 19 additional associations (Supplementary Data 8). For example, the variants rs3217893_C > T and rs2476601_A > G, fine-mapped for type 2 diabetes and rheumatoid arthritis (RA) in FinnGen respectively, were found to have associations with medications gliclazide, which is a sulfonylurea

used in the treatment of Type 2 diabetes, and steroid prednisolone which can be used to treat RA, only when the genetic correlation information was included.

Query variants were often associated with multiple UKBB traits (median 5) that reflected related diseases and medications (Supplementary Data 8). For instance, rs11591147_G > T, a missense variant of PCSK9, identified as a disease-causal variant in FinnGen for statin medication was found associated with the UKBB traits related to different statin medications along with several cardiovascular traits. Less commonly, we found evidence for causal association of variants to seemingly unrelated traits. For example, rs9349379_A > G, an intron variant and eQTL for PHACTR1, identified by fine-mapping the FinnGen primary trait−triptan, which is a medication used to manage migraine, was found to be associated with several UKBB traits related to migraine such as the phenotype itself, migraine medications such as suma-triptan, ibuprofen and paracetamol and also the presence of family history. However, we also found associations with angina, myocardial infarction and ischaemic heart disease, with the migraine-protective allele acting as a risk factor for cardiovascular traits. This matches results from a Mendelian randomisation study of migraine and cardiovascular disease[16] but is in contrast to observational studies where migraine is considered positively associated with cardiovascular traits[17]. Such discrepancies between genetic and observational studies in other traits have often been resolved in favour of the genetic result, through the identification of some confounding factor which led the observational studies to report inverse relationships, and it has been suggested that certain non-triptan migraine therapies might act to

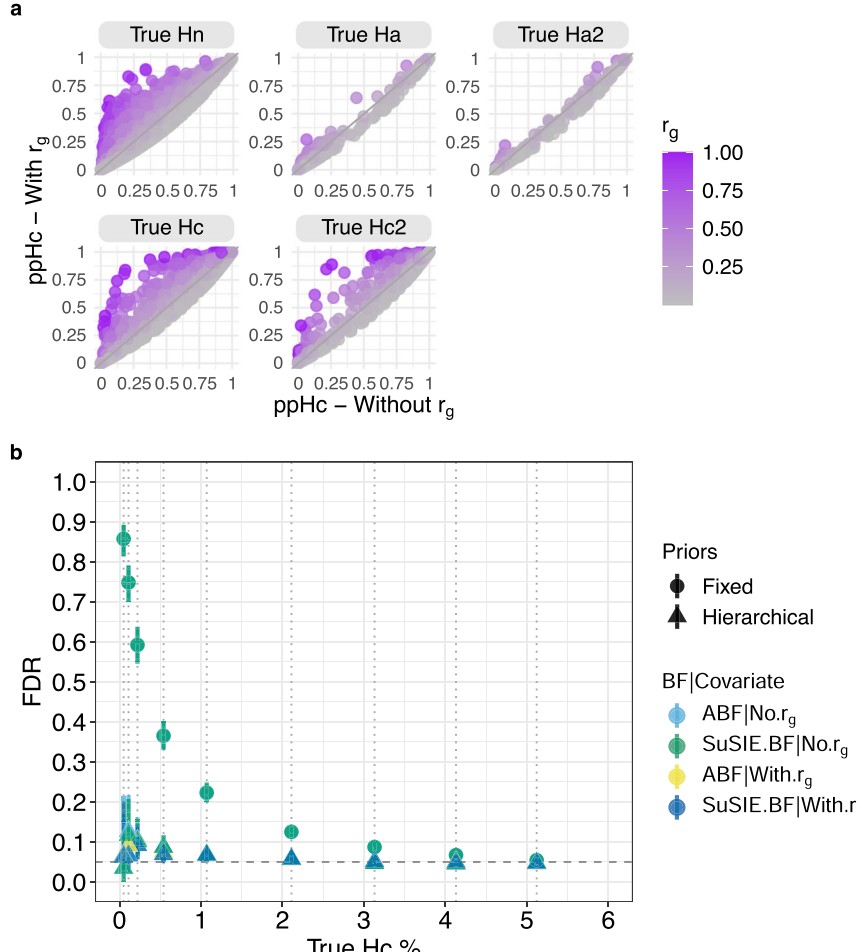

**Fig. 3 | Effects of covariate inclusion and varying proportions of simulated hypotheses. a** Comparison of the posterior probability of Hc (ppHc) obtained with (*y*-axis) and without (*x*-axis) the inclusion of genetic correlation ($r_g$) in the hierarchical model (using ABF). The panels represent the traits of different simulated hypotheses (Methods). **b** The proportion of Hc traits was varied as shown in the *x*-axis (dotted vertical lines), to compare Hc predictions using the fixed and hierarchical priors with different BF, both with and without the inclusion of the genetic correlation ($r_g$) covariate. The number of true Hn and Ha traits were maintained the same at 88048 and 4700 respectively. The *y*-axis represents the estimated FDR−the proportion of traits assigned as Hc in each dataset which were simulated as Hn or Ha with 95% confidence intervals (dashed line − 0.05 FDR).

increase cardiovascular risk[16]. However, this pleiotropy did not appear at another migraine-identified variant, rs11172113_T > C, an intronic variant of *LRP1*, which was fine-mapped for the same FinnGen primary trait of migraine and found to be independently associated with several migraine-related phenotypes in UKBB but not with any of the cardiovascular traits (Fig. 5).

Other examples of pleiotropic variants include rs2476601, a non-synonymous variant in *PTPN22* which we found to be causally associated with multiple autoimmune diseases and their treatments as well as skin cancer, with the autoimmune-protective allele increasing risk of cancer (Supplementary Fig. 9). We also found a complex set of associations with two variants in *APOE*, rs429358 and rs7412 that jointly define the three major structural isoforms of APOE[18], ε4, ε3 and ε2 (Supplementary Fig. 10). ε2 represents the TT haplotype corresponding to the rs429358 and rs7412 variants, ε3 is represented by TC and ε4 by the CC haplotype[19]. We found associations with increased risk of Alzheimer's disease, statin medication, angina and ischaemic heart disease with the ε4 allele with reference to the ε2/ε3 genotype. We also found a protective effect of ε4 compared to ε2/ε3 on traits related to a family history of diabetes and blood pressure which correspond to similar traits found in FinnGen as well as a protective effect of ε3/ε4 compared to ε2 for deep venous thrombosis might be related to the

ε3/ε4 genotype with reference to ε2 and might indicate the ε2 allele. These findings align with previous studies on disease associations with different *APOE* genotypes[20] and highlight the ability of SuSiE to map traits to distinct alleles in LD.

**Individual variant analyses**

CoPheScan can also be used to study single variants if sensible prior values can be supplied. We considered exemplar non-synonymous variants in two genes, *TYK2* with established allelic heterogeneity and associations to multiple immune-mediated diseases, and *SLC39A8*, with established pleiotropic function. We ran CoPheScan with SuSiE BF and priors inferred from the disease-causal variant analysis above ($p_a \approx$ 3.82e-5 and $p_c \approx$ 1.82e-3), considering as query traits 2275 UKBB and 56 additional traits potentially related to either gene from the GWAS Catalog (Supplementary Data 3).

*TYK2* which encodes the tyrosine kinase 2 enzyme has multiple missense variants that have been associated with a range of immune-mediated diseases (Supplementary Data 11). We considered four: rs35018800_G > A (MAF: 0.0082), rs34536443_G > C (MAF: 0.0465), rs12720356_A > C (MAF: 0.0979), and rs55882956_G > A (MAF: 0.0017). rs35018800_G > A and rs55882956_G > A with the lowest MAF showed no association with any trait. rs34536443_G > C was associated with 3

UKBB and 5 GWAS Catalog traits, all immune-related and previously established associations, including psoriasis, RA, JIA (Juvenile Idiopathic Arthritis), Type 1 DM, and hypothyroidism. The variant rs12720356_A > C was associated with ulcerative colitis, psoriasis,

Crohn's disease, SLE (Systemic Lupus Erythematosus) and RA traits from the GWAS Catalog, but not with any of the UKBB traits (Fig. 6).

The highly pleiotropic variant, rs13107325_C > T, of *SLC39A8* (solute-carrier family gene which encodes the ZIP8 protein), was associated with 14 UKBB and 3 GWAS Catalog phenotypes, replicating several known associations[21] with hypertension, schizophrenia, Crohn's disease, urinary incontinence, musculoskeletal system-related traits such as osteoarthritis and traits related to alcohol dependence.

We used this region to perform a sensitivity analysis, selecting four variants—rs6855246, rs35225200, rs35518360, rs13135092, in LD with rs13107325_C > T ($r^2 = 0.816$–$0.943$) and running CoPheScan as if each had been selected as the causal variant. This allows us to explore two related questions: either, to what extent can two causal variants in LD cause false positive findings, or, to what extent CoPheScan might still detect an association if the "causal" variants supplied to CoPheScan are not really causal, but in LD with the causal variant. We found that CoPheScan was indeed sensitive to this misspecification, where out of the 17 traits identified as causally associated with rs13107325, 4 had ppHc < 0.6 with rs13135092 ($r^2 = 0.943$) and 11 with rs6855246 ($r^2 = 0.816$). The results were increasingly discrepant as the $r^2$ with rs13107325_C > T decreased (Fig. 7 and Supplementary Fig. 11). The group of traits with high ppHc across multiple variants tended to have larger minimum *p*-values in the region compared to those for which ppHc was low across multiple variants, suggesting that CoPheScan will be best at discriminating between potential causal variants in LD when the association signal in the query data is strong.

Finally, we sought to verify previously proposed causal associations between the *HMGCR* variant rs12916_T > C and metabolic traits. *HMGCR* encodes HMG-CoA reductase which is targeted by statins to lower LDL cholesterol. Previously, *HMGCR* variants have been used as a proxy for statin effect to show a higher risk of type 2 diabetes and body mass index (BMI) in MR studies[22]. But the validity of this has been challenged with evidence that there may be distinct causal variants underlying type 2 diabetes, BMI and HMGCR levels[23].

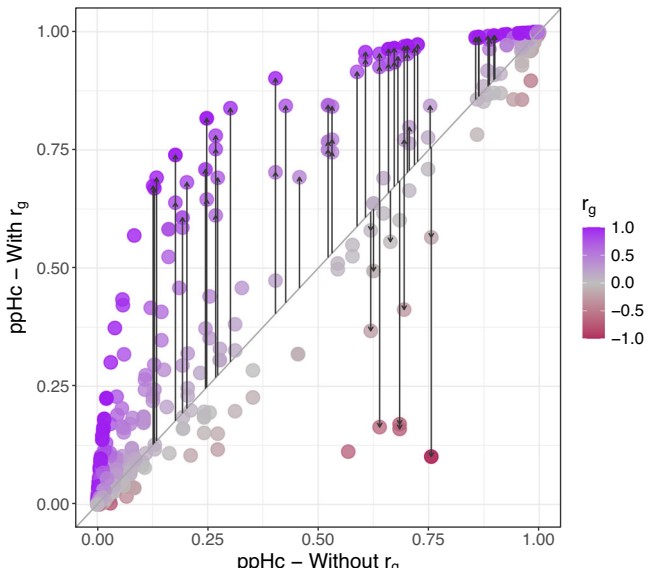

**Fig. 4 | Genetic correlation detects additional phenotypes.** Hierarchical models of the FinnGen/UKBB dataset with/without genetic correlation ($r_g$). The posterior probability of Hc (ppHc) of traits with and without the inclusion of genetic correlation ($r_g$) are shown on the *y* and *x* axes respectively. The arrows represent the traits which show a difference of >0.1 ppHc after inclusion of $r_g$ (compared to the model without) and also have a ppHc > 0.6. The traits are coloured to represent their $r_g$ with the primary trait.

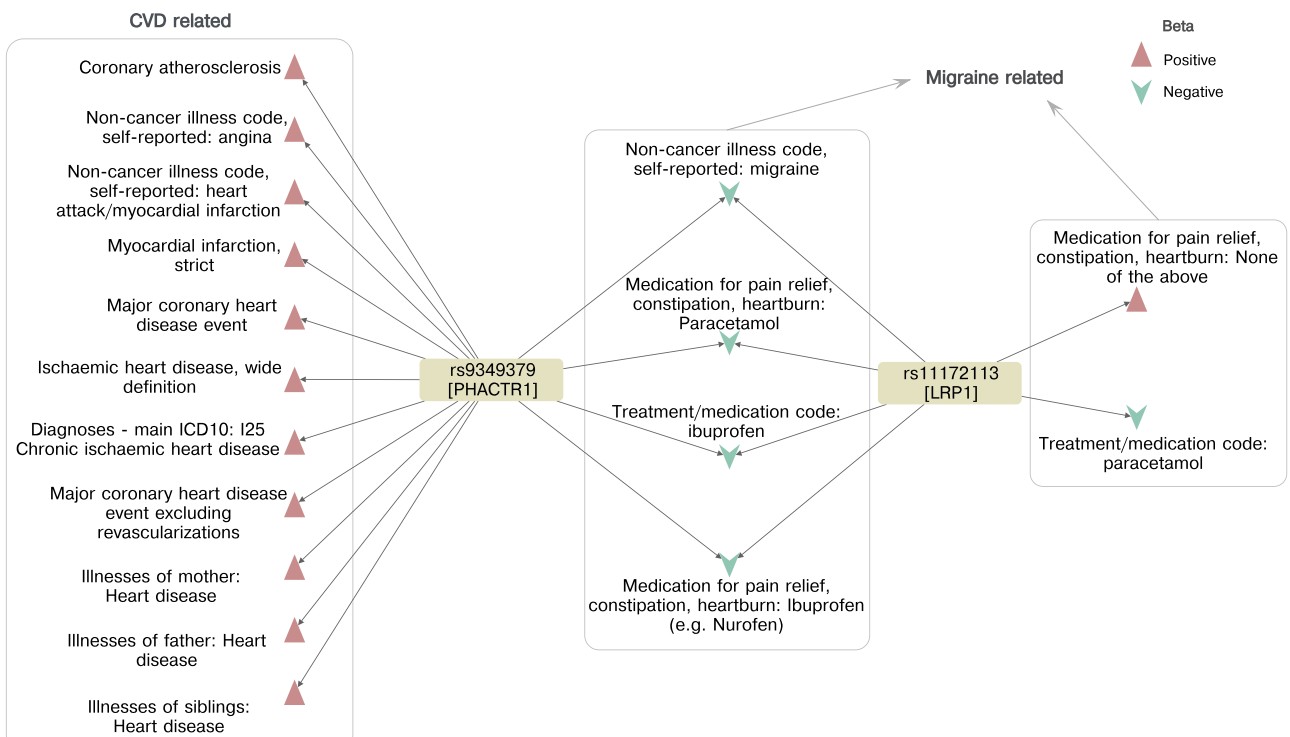

**Fig. 5 | Causal associations of Migraine related variants.** Hc associations (ppHc > 0.6) of the PHACTR1 variant rs9349379_A > G and rs11172113_T > C, a LRP1 variant. The direction of effect (beta) is shown with respect to the G and C allele respectively.

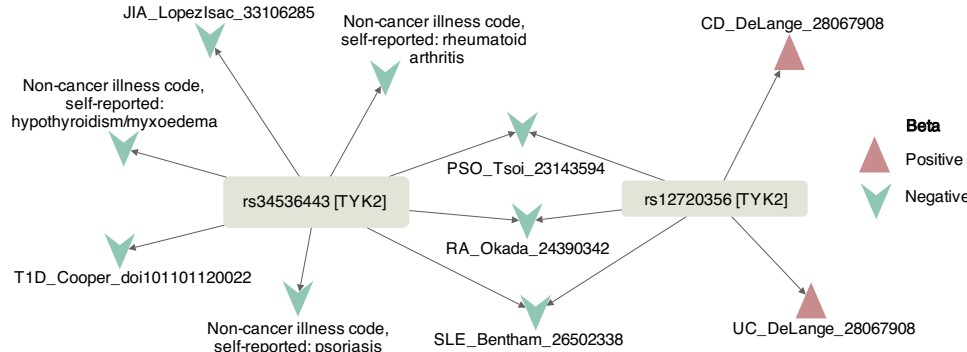

**Fig. 6 | CoPheScan analysis of a gene with allelic heterogeneity: *TYK2*.** Plots showing Hc associations (ppHc > 0.6) of *TYK2* variants rs34536443_G > C and rs12720356_A > C. The direction of beta is shown with respect to the ALT allele (C in both cases). T1D Type 1 Diabetes Mellitus, JIA Juvenile Idiopathic Arthritis, PSO Psoriasis, RA Rheumatoid Arthritis, SLE Systemic Lupus Erythematosus, CD Crohn's Disease, UC Ulcerative Colitis.

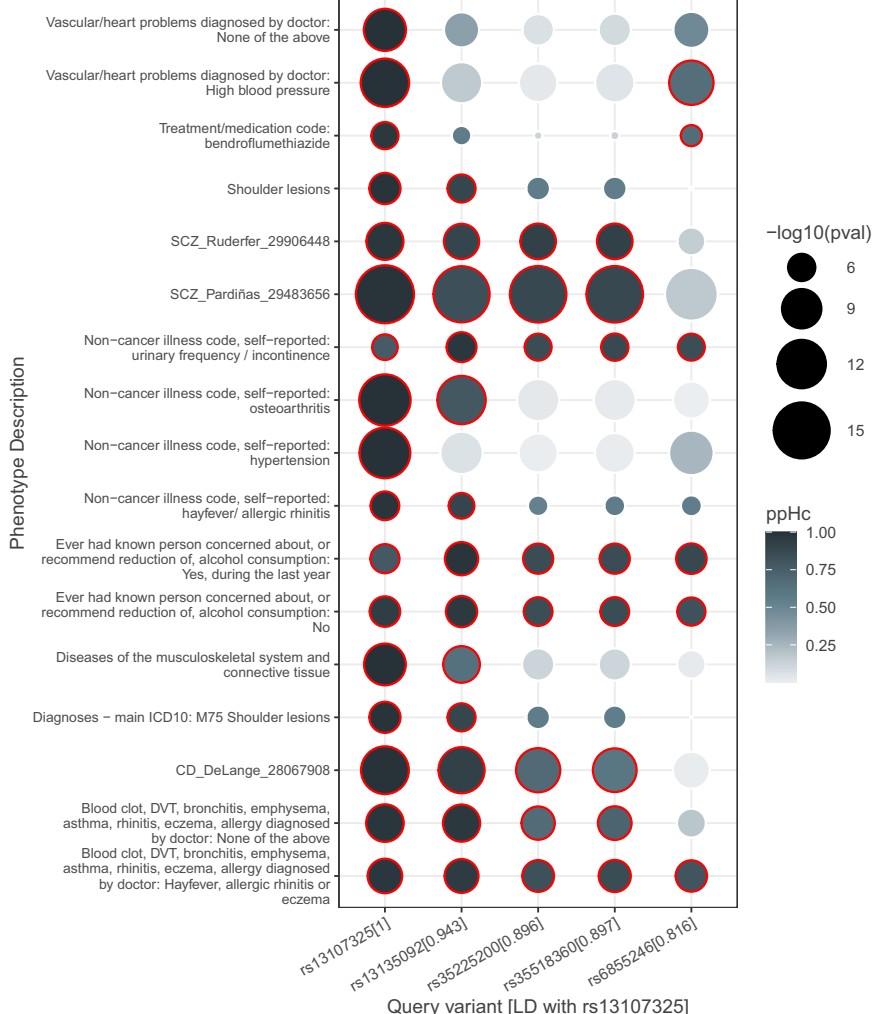

**Fig. 7 | Sensitivity analysis with SLC39A8 variants.** Heatmap of ppHc of *SLC39A8* variants with LD $r^2 > 0.8$ with the rs13107325_C > T (bottom row) variant. The points are scaled based on their unadjusted -log10(pval) as given in the input summary GWAS data and a red border indicates a ppHc of >0.6. CD Crohn's disease, SCZ schizophrenia.

We performed CoPheScan analysis on the UKBB traits: LDL, BMI, type 2 diabetes, waist circumference and weight. We identified a known causal association with LDL (ppHc = 1). Despite significant observed *p*-values at rs12916 at BMI, weight and waist circumference, CoPheScan consistently concluded that while the region contained a causal variant for each trait, it was not rs12916 (ppHa > 0.99). In fact, no credible sets were identified in the *HMGCR* gene region and the

SuSIE signals from these traits indicate the presence of an alternative causal *POC5* variant (Supplementary Fig. 12). This implies that genetic studies that demonstrated a relationship between statin therapy and BMI/T2DM through *HMGCR* variants as a proxy might be incorrect[23] as they studied the SNPs in isolation while ignoring their regional context[24]. CoPheScan is thus valuable in verifying assumptions in instrumental variable analyses.

## PheWAS of protein-associated variants

One challenge of GWAS has been to link disease associations to their causal genes. PheWAS allows us to start with variants with known causal function on a protein and ask which diseases are also causally associated, exploiting the low false positive rate of CoPheScan. We began with 505 plasma protein QTLs[25] identified as single variant credible sets in fine-mapping of 527 plasma proteins. Nine variants were identified to be associated with UKBB traits (Table 1 and Supplementary Data 9). Among the established associations, we found an association between a pQTL for APOC1 and high cholesterol, as well as reported treatment with the cholesterol-lowering simvastatin. Both associations make sense given the known biology of APOC1, but only the first would have been detected in scanning for significant $p$-values, as the $p$-value for high cholesterol at this SNP ($p = 6.19 \times 10^{-19}$) is much lower than for simvastatin ($p = 9.59 \times 10^{-4}$), emphasising the value of exploiting the additional information that we believe the variant to have a causal effect on a measurable phenotype (Supplementary Fig. 13).

We also found a novel association for rs214830_G > C, a pQTL for TGM3 which was associated with skin cancer (ppHc = 0.75). TGM3 is required for skin development and is normally expressed in the spinous/granular layers of the epidermis. Its expression was found to be absent in melanoma and squamous cell carcinoma of the skin but strongly expressed in basal cell carcinoma (BCC), suggesting it could be a specific marker for BCC diagnosis[26]. Association of variants in *TGM3* with BCC have also been reported[27–29] but rs214830_G > C was not always the top variant and GWAS associations can mark causal effects in neighbouring genes[30]. Our analysis suggests this association could be directly causal, with TGM3 involved in the development of BCC as well as acting as a biomarker.

Finally, we considered 3586 variants labelled as protein-truncating (PTV) in the UKBB summary data with MAF > 0.001, consisting of those predicted by VEP to be stop_gained, frameshift, splice_acceptor and splice_donor. The fraction of query variants that were found to be causally associated with at least one trait in UKBB was much lower for PTV (~0.31%) than for disease-causal variants identified in FinnGen (~40%) and pQTL (~1.8%) (Table 2, Supplementary Figs. 5 and 6).

Examination of the Markov chain Monte Carlo (MCMC) chains showed issues with mixing for the PTV example which were not seen with the other datasets (Supplementary Fig. 5). When we examined the inferred priors (Supplementary Data 12) obtained from this model, we observed that the $p_c/p_a$ ratio was ~1.02, indicating that the inferred $p_a$ and $p_c$ priors were almost the same. Our PTV consisted of four VEP classes, but while the MAF distribution of the stop-gained PTV was similar to missense variants, those of the other PTV (frameshift, splice donor and splice acceptor) were similar to synonymous variants (Supplementary Fig. 14a). As selection can constrain MAF, we hypothesised that the VEP stop_gained class might be more enriched for functional variation than the set of four classes we had used. We considered two ways to enrich the PTV set for functional variation: either using just this subset of the stop-gained PTVs or using the PTVs which were also defined as high confidence homozygous predicted loss-of-function (pLoF) variants in gnomAD[31]. pLoF were predominantly rare, such that the pLoF subset of PTV variants had a higher number of rare variants compared to the stop-gained subset (Supplementary Fig. 14b).

We ran the hierarchical models for these two subsets of PTVs (Supplementary Fig. 15). Comparing the priors (Supplementary Data 12) across the different datasets tested we observed that the ratio of prior probabilities for the query variant or a non-query variant to be causal, $p_c/p_a$ (Table 2) obtained using the pLoF variants (2.59) was second only to the ones obtained using the FinnGen disease-related variants. The ratio from the stop-gained variant model (1.39) was similar to the pQTL variant model (1.28). This shows that sets of query variants which have a higher functional enrichment are expected to have a high $p_c/p_a$ ratio.

26 associations were identified using all the PTV variants. All 15 associations detected with the stop-gained PTVs and 7 from the pLOF overlapped with those from the whole set. Of the combined 26 PTV-trait associations (Supplementary Data 10), many corresponded to known effects. One of them is, rs2066847_G > GC, a *NOD2* frameshift mutation, which is reported as a pathogenic variant for inflammatory bowel disease in ClinVar and was associated with several phenotypes related to Crohn's disease and mouth ulcers in our analysis (Supplementary Fig. 16). However, as seen with migraine and cardiovascular disease above, the association with mouth ulcers occurs in the opposite direction to the established comorbidity of Crohn's disease and mouth ulcers in the population, with the Crohn's disease risk allele appearing protective for mouth ulcers. Note that in the mouth ulcer

### Table 1 | Hc associations detected with pQTL variants

| Query variant | Protein | Direction | UKBB traits detected as Hc |
|---|---|---|---|
| rs11591147 | PCSK9 | risk | High cholesterol, cholesterol-lowering medication, ischaemic heart disease |
| rs5743618 | TLR1 | protect | Asthma |
| rs3775291 | TLR3 | risk | Hypothyroidism/myxoedema |
| rs3136516 | F2_Prothrombin | risk | Venous thromboembolism |
| rs34324219 | TCN1 | protect | Pernicious anaemia |
| rs964184 | APOC3 | risk | Cholesterol-lowering medication |
| rs116843064 | ANGPTL4 | risk | High cholesterol |
| rs5112 | APOC1 | risk | High cholesterol, cholesterol-lowering medication |
| rs214830 | TGM3 | risk | Skin cancer |

### Table 2 | Summary of tested variants and phenotypes from real data

| Query variant set | N QV | N QT | N QV-QT pairs* | N QV-QT detected as H_c | N (%) unique variants detected as H_c | pc/pa |
|---|---|---|---|---|---|---|
| FinnGen (with $r_g$) | 75 | 1582 | 130697 | 184 | 30 (40%) | 95.6 |
| FinnGen (no $r_g$) | 136 | 2275 | 193706 | 328 | 43 (31.62%) | 47.5 |
| pQTL (no $r_g$) | 505 | 2275 | 954616 | 29 | 9 (1.78%) | 1.28 |
| PTV all (no $r_g$) | 3586 | 2275 | 4359271 | 26 | 11 (0.31%) | 1.02 |
| PTV gnomAD (no $r_g$) | 366 | 2275 | 292787 | 7 | 2 (0.54%) | 2.59 |
| PTV stop gained (no $r_g$) | 911 | 2275 | 837060 | 15 | 6 (0.66%) | 1.39 |

QV query variant, QT query trait, N QV-QT number of trait-variant associations. The number of QT were lower for the FinnGen/UKBB dataset for the 'with $r_g$' case as only traits having data with the primary traits available were retained (Methods).

trait, the effect sizes were opposite in two other SNPs identified as a credible set in SuSiE analyses of both traits (Supplementary Fig. 16).

## Discussion

Detection of pleiotropic effects of genetic variants is an essential component of target discovery and drug repositioning. PheWAS typically takes information from marginal statistics at query variants in isolation of their neighbours, which can lead to false positives when multiple causal variants exist in LD. CoPheScan considers not only how small a $p$-value is at a given variant, but how small it is in comparison to its neighbours, and estimates how much upweighting should be applied due to the information that the variant is in a query variant set. In our simulations, CoPheScan showed considerably better control of false positive calls compared to a standard PheWAS approach, at the cost of lower sensitivity where multiple causal variants exist in a region. Whilst the higher false positive rate for standard PheWAS testing can be mitigated by the use of a second-stage analysis testing for colocalisation, that is not possible in the case of query SNPs selected for their known effects on a protein, such as the PTV considered here.

CoPheScan learns how much to upweight query variants through the prior parameter $p_c$ and the ratio of average $p_c$ to average $p_a$ is a useful measure of enrichment of causal variants for the set of query traits amongst the set of query variants. This measure can be used to assess the quality of any choice of variant set, with values close to 1 indicating a weak choice. It may vary considerably across query variant sets for the same set of query traits, as seen in the PTV analyses. However, while restriction to a smaller set of query variants with greater enrichment is likely to find a higher proportion of causal associations with the smaller set, this will not necessarily enhance discovery: whilst the majority of the discoveries found using the smaller, more enriched sets of PTV were also found in the larger unfiltered set, this restriction also meant losing plausible discoveries that didn't fall into either of the more restricted classes.

We allow $p_c$ to vary between variants by exploiting additional external information in a regression framework. In our disease-variant focused analysis, we used the genetic correlation between index and query traits, but this could also be a categorical variable, such as the predicted deleteriousness of a missense variant, or the level of evidence for the functional effect of a PTV. Our model can exploit covariate information that relates to query trait-query variant pairs, but would need to be extended to accommodate other information. For example, we might see modest evidence for causal association of a medication trait with a given query variant, but intuitively trust the result is true because of stronger evidence at the same variant with the disease itself. The difference in inference in such a case might be explained by the smaller numbers of individuals reporting use of a specific medication. We could consider exploiting genetic correlation between query traits by using a multivariate prior, with covariance linked to the genetic correlations. While this is beyond the scope of the current study, we hope our use of covariate-informed priors illustrates the potential for external information to be exploited when conducting PheWAS and other genetic studies.

While the simulations emphasised the importance of learning $p_c$ in a hierarchical model for accurate inference, point estimates can be substituted if required. This borrowing of priors from a larger dataset is beneficial in scenarios where we might want to use CoPheScan to test associations between a small set of variants and phenotypes, as running a hierarchical model on limited data will not result in optimal prior estimates. However, we strongly advise that careful consideration is needed to ensure the larger dataset in which the priors are learnt is a good match for the limited dataset under consideration.

One of the advantages of incorporating SuSiE in CoPheScan is the ability to detect allelic heterogeneity at a locus. We demonstrated this with two well-known distinct variants in the *TYK2* gene which were associated with overlapping sets of immune-mediated disorders. This analysis also highlighted the importance of surveying disease-specific GWAS studies and not relying solely on biobanks which may hold relatively low numbers of cases of any individual disease. For example, only three UKBB traits showed any association compared to seven of our curated immune-mediated disease GWAS, and while psoriasis in UKBB (4192 cases) was identified with one variant, psoriasis in Tsoi's GWAS study (10558 cases) was identified with two. While biobanks remain incredibly useful for common traits such as cardiovascular and metabolic diseases, carefully curated bespoke GWAS of less common traits should be included in any PheWAS to complement the biobank resources and reveal the full spectrum of pleiotropy. This is particularly important because predicted beneficial effects of targeting a protein may be countered by on-target side effects on other traits, as we saw where the autoimmune-protective variant in *PTPN22* was associated with an increased risk of skin cancer.

Our CoPheScan approach has some specific limitations. The signals obtained from the multiple variant assumption rely on the available LD information. Zou et al. demonstrated that the performance of SuSIE degrades when presented with out-of-sample LD matrices[14]. In cases where in-sample LD matrices are not accessible, it is recommended to utilise out-of-sample LD from large reference panels. Our experience with SuSIE is that as the number of causal variants increase, the performance (ability to detect all causal variants and/or ability to detect the correct causal variants) may decrease. In the case that a true causal variant signal is missed for our query variant, as occurred in around 35% of our simulations with two causal variants, CoPheScan concludes Ha−i.e. a false negative. SuSiE is likely to miss a higher fraction of secondary causal variants when the true number of causal variants increase, which would be expected to lead to greater false negatives for CoPheScan. Importantly, we do not see any increase in false positives when simulating two compared to one causal variant. Analysis of rare events in large samples with standard methods can cause bias in regression summary statistics. Here, we used careful QC, thresholding on the number of events / MAF, but a better approach would be to use methods specifically developed to deal with this such as REGENIE[32] to generate input to CoPheScan. The current form of CoPheScan only allows single-ancestry studies which will be addressed in future iterations and allow an increase in power to detect rare variants.

GWAS causal variants, even when identified with confidence, remain challenging to interpret partly because it can be hard to link them with confidence to their causal genes. Protein-altering variants have thus become increasingly important because their function on a gene is presumed known. The different relative enrichments in different sets of PTV we ran suggests that incorporating external evidence on the plausibility of a putative PTV having a functional effect will increase accuracy in PheWAS of these variants. However, as highlighted here, they often have very low minor allele frequencies. Thus, larger biobanks are still needed both for analysis of less common traits with common variants and for analysis of rare functional variants. It is thus encouraging that UKBiobank and FinnGen studied here are complemented by the Japan Biobank[33], the Million Veteran Program[34] and the Uganda Genome Resource[35], which should allow CoPheScan, together with efforts at multiple ancestry fine mapping[36], to reveal more completely the pleiotropic spectrum of protein-altering genetic variation.

## Methods
### Simulated data

We simulated case-control summary statistics using the EUR samples in the 1000 Genomes phase 3 reference data[37]. LD-independent blocks were identified using lddetect[38] and haplotypes containing 1000 SNPs with MAF > 0.01 were extracted from the reference data[10,39]. We used simGWAS[39] to simulate summary statistics with either one or two

causal variants for the corresponding LD blocks with 10000 cases and 10000 controls.

We simulated GWAS summary statistics for 110,000 traits to evaluate hypothesis discrimination, with all genomic regions containing 1000 SNPs. We sampled query causal variants at random from the 1000 SNPs and simulated each trait to correspond to one of the three hypotheses. The traits within the simulated dataset were divided based on the number and position of the causal variants within their genomic region:

1. **True Hn**: No causal variants within the genomic region.
2. **True Ha**: A trait with a single causal variant that is not the query variant.
3. **True Ha2**: A genomic region simulated with two distinct causal variants, none of which are the query variant.
4. **True Hc**: A trait with a single causal variant that is the same as the query variant.
5. **True Hc2**: Two distinct causal variants, where one of them is the same as the query variant.

The 110000 simulated traits were comprised of 88048 true Hn, 6276 each of true Hc and Hc2, and 4700 each of Ha and Ha2 traits. We also simulated genetic correlation values for each of these traits where the Hc traits were assigned a higher proportion of high $r_g$ values when compared to the Hn and Ha traits (Supplementary Fig. 2).

We used conventional PheWAS approaches, based on selecting associations that cross a threshold $p$-value after accounting for multiple testing. We used the Benjamini and Hochberg method (BH) to control the FDR < 0.05 which corresponded to a $p$-value < 7.5e-3 and Bonferroni correction with a $p$-value < 4.55e-7 to control the family-wise error rate (FWER) at 0.05. In parallel, we used different approaches of CoPheScan to analyse this dataset:

1. Fixed priors and Approximate Bayes Factors (ABF).
Fixed priors used with the simulated data were adapted from coloc (described in the Supplementary Methods) where $p_a \approx 0.81$ and $p_c \approx 0.091$.
2. Fixed priors and SuSIE Bayes factors
3. Hierarchical priors, ABF, with and without genetic correlation ($r_g$)
4. Hierarchical priors, SuSIE Bayes factors, with and without $r_g$

The hierarchical model of CoPheScan was run for 3e5 iterations for the models without $r_g$ and 1e6 iterations for the ones with $r_g$ and the chain was thinned by retaining every 30th and 100th observation respectively (Supplementary Fig. 3). The first 50% of the remaining 1e4 observations were discarded and the average prior ($p_n$, $p_a$, $p_c$) and posterior probabilities ($ppH_n$, $ppH_a$, $ppH_c$) were calculated.

Therefore, the output of each analysis for each trait was summarised by the Hc hypothesis when $ppH_c > 0.6$, and Hn when $ppH_n > 0.2$ and Ha for the remaining traits. When multiple signals were detected by SuSIE in the same genomic region, CoPheScan was run on each of them. Here, we assigned the hypothesis to each signal as the thresholds specified above. The first hypothesis to occur in the ranking order of Hc, Hn, and Ha was assigned to the trait, i.e., when there was at least one signal which was assigned as Hc, the trait was taken to be Hc. Next, any signal with Hn but no Hc was assigned as Hn, because we wanted to be conservative in calling Ha which might rule out a pleiotropic effect. In the absence of a Hc and Hn signal, we checked for the presence of Ha and where there were multiple Ha signals we report the minimum Ha of all the signals.

We also simulated datasets where we varied the number of true Hc traits {50, 100, 200, 500, 1000, 2000, 4000, 5000} while maintaining the true Hn and Ha traits the same at 88048 and 4700 respectively. The percentage of Hc traits in the datasets corresponded to {0.05%, 0.11%, 0.22%, 0.54%, 1.07%, 2.11%, 3.13%, 4.13%, 5.12%}.

### Disease causal query variants from FinnGen
We downloaded SuSIE[13,14] fine-mapping results from FinnGen[40] release R5, which has a sample size of 218,792 with 2,803 endpoints (https://www.finngen.fi/en/access_results). For each endpoint, we filtered variants that belonged to a credible set of size 1 and were also present in the UKBB dataset. We retained 136 variants, fine-mapped from 141 FinnGen traits, for further analysis. 69 out of these 141 FinnGen primary traits, had closely matching UKBB traits with pre-computed genetic correlation data. Thus, 75 variants from these matching traits were used for the hierarchical model using $r_g$ (Supplementary Data 4 and 5).

### pQTL variants
We downloaded summary data from a GWAS of plasma protein levels measured with 4,907 aptamers (corresponding to 4719 proteins) in 35,559 Icelanders from Ferkingstad et al.[25]. We fine-mapped the region around each signal under a single variant assumption. This is equivalent to taking only the first signal in a stepwise fine-mapping procedure. We made this conservative choice to address the lack of access to an LD matrix for the Icelandic population, making it difficult to trust secondary signals found by stepwise regression or other multiple causal variant methods such as SuSIE. We obtained 505 SNPs associated with 527 proteins for testing associations with the UKBB phenotypes (Supplementary Data 6).

### Protein truncating variants
We selected 3586 protein truncating variants (PTV) with MAF > 0.001 (Supplementary Data 7) from the UKBB variants (https://broad-ukb-sumstats-us-east-1.s3.amazonaws.com/round2/annotations/variants.tsv.bgz), which were annotated as frameshift (883), stop gained (911), splice acceptor (682) and splice donor (1110) variants, using VEP[41] (The Ensembl Variant Effect Predictor, 85).

We also downloaded homozygous pLoF from gnomAD (v2.1.1, https://gnomad.broadinstitute.org/downloads). Out of these, we selected 366 variants, which were classified as either 'lof' or 'likely_lof' and were in common with the 3586 UKBB PTV variants[31].

### Individual variant analyses
We chose four *TYK2* variants: rs35018800, rs34536443, rs12720356 and rs55882956; a *SLC39A8* variant, rs13107325; and a *HMGCR* variant, rs12916, to examine the performance of CoPheScan for region-specific analyses[42-44].

### Query phenotypes
We used 2275 phenotypes from the UK Biobank (http://www.nealelab.is/uk-biobank). We obtained in-sample linkage disequilibrium matrices from https://registry.opendata.aws/ukbb-ld/[15]. We included all the 2275 traits in the CoPheScan analysis of the FinnGen, pQTL and PTV variants.

We downloaded genetic correlation[45] data between UK Biobank traits and disorders estimated using LD score regression[46] from https://ukbb-rg.hail.is/. In the FinnGen/UKBB dataset, only 1582 out of the 2275 traits had genetic correlation ($r_g$) estimates with the UKBB traits mapped to the FinnGen primary traits. So, only these traits were used for the hierarchical model that included $r_g$. Additionally, we checked the allele counts (AC) of the query variant in the phenotype files and only retained the QV-QT pairs for association testing when the AC in the cases >25. After reviewing the results, to increase stringency, we further removed QV-QT pairs, identified as being causally associated with AC < 30 to reduce false positive detection. Individual results presented in tables have been trimmed to reflect this more stringent criterion (removing five Hc results) but estimates directly from models (eg pa/pc) include observations with AC between 26 and 30.

For the phenome-wide scan of the *TYK2* and *SLC39A8* variants, we downloaded 56 additional publicly available GWAS summary statistics

of phenotypes related to immune-mediated and psychiatric diseases (Supplementary Data 3)[47–49]. In the case of the *HMGCR* variant we used additional quantitative UKBB phenotypes: LDL direct, Body mass index (BMI) and Weight. We used UKBB LD matrices for data from European populations, and for other populations, we extracted LD from the 1000 genomes phase 3 reference data[37].

The lists of phenotypes used with the different variants are provided in Supplementary Data 2 and 3. We used Phase II HapMap[50] obtained from (https://ftp.ncbi.nlm.nih.gov/hapmap/recombination/2011-01_phaseII_B37/) to subset regions from the summary statistics data around the query variants. We excluded variants in the HLA region (20MB–40MB) from the analysis.

Visualisations of the causal trait-variant associations identified with CoPheScan were done using Cytoscape[51] 3.9.0.

### Functional annotation

Previously reported variant/gene associations with diseases were obtained from the Open Target Platform and Open Target Genetics[48,52] https://platform.opentargets.org/downloads. We used the DrugBank online resource (https://go.drugbank.com/drugs/) for indications of medications that were associated with the variants[53].

### Reporting summary

Further information on research design is available in the Nature Portfolio Reporting Summary linked to this article.

## Data availability

The simulated summary statistics and processed files are available on figshare (https://doi.org/10.6084/m9.figshare.24939408)[54]. UKBB summary statistics are available at http://www.nealelab.is/uk-biobank; Phase II HapMap were downloaded from https://ftp.ncbi.nlm.nih.gov/hapmap/recombination/2011-01_phaseII_B37/; UKBB in-sample LD matrices are available at https://registry.opendata.aws/ukbb-ld; SuSIE fine-mapping results from the FinnGen Freeze 5 cohort are publicly available at https://www.finngen.fi/en/access_results; GWAS Catalog summary statistics were downloaded from https://www.ebi.ac.uk/gwas/; GWAS of plasma protein levels used for pQTL fine-mapping are publicly available at https://www.decode.com/summarydata/. Source data are provided with this paper.

## Code availability

The CoPheScan R package is available on CRAN at https://cran.r-project.org/package=cophescan and on Zenodo (https://doi.org/10.5281/zenodo.11654394)[55]. A shiny app to browse the results is available here: https://ichcha-m.shinyapps.io/cophescan-app/, the code for which can be found at https://github.com/ichcha-m/cophescan-app. The code to reproduce the simulated summary statistics and processed datasets are available here: https://github.com/chr1swallace/cophescan-manuscript-sim-summary-data and https://github.com/ichcha-m/cophescan-paper.

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

## Acknowledgements

This work is funded by the Wellcome Trust (WT220788), the MRC (MC_UU_00002/4), G.S.K. and M.S.D. and supported by the NIHR Cambridge BRC (BRC-1215-20014). The views expressed are those of the author(s) and not necessarily those of the NHS, the NIHR or the Department of Health and Social Care. This research was funded in whole, or in part, by the Wellcome Trust (WT220788). For the purpose of Open Access, the author has applied a CC BY public copyright licence to any Author Accepted Manuscript version arising from this submission. We want to acknowledge the participants and investigators of the FinnGen and UK Biobank study, and the Neale lab for publicly sharing UK Biobank summary statistics.

## Author contributions

C.W. conceived the project, designed the experiments and supervised the research. I.M. performed the experiments. C.W. and I.M. performed statistical analysis and developed the software. C.W., I.M., A.C., J.H.S., S.L. and M.K.S. contributed to the experimental design. G.R. contributed analysis tools. C.W., I.M., A.C., J.H.S. and S.L. analysed the data and revised the manuscript.

## Competing interests

A.C. is a full-time employee of GSK. S.L. is a full-time employee of Merck Sharp & Dohme LLC (MSD), a subsidiary of Merck & Co., Inc., Rahway, NJ, USA. C.W. and I.M. receive funding from GSK and MSD. C.W. is a part-time employee of GSK, but this research was conducted within her academic role. J.H.S. and M.K.S. were full-time employees of MSD during their involvement in the project. The remaining authors declare no competing interest.
