## [Peer Review File · Nature Communications]

CoPheScan: phenome-wide association studies accounting for linkage disequilibriumREVIEWER COMMENTS

Reviewer #1 (Remarks to the Author):

I read with interest this paper by Manipur and colleagues that describes a method to query large scale GWAS type data for colocalisation with a variant associated with a trait of interest. This is a commonly asked question, often answered in very approximate ways, and this work provides an elegant and well motivated approach to answer the question. So there is absolutely no doubt in my mind that this paper should be published- this looks very useful and should receive a lot of attention/usage. My comments mostly relate to clarifications and presentation, which I think could help readers like myself.

My first request relates to better spelling out the methodological difference between the commonly used coloc methods (either as in Giambartolomei et al, or the SuSiE version) and this new algorithm. As I understand it, there are three modifications: (i) adapting coloc to a known causal variant situation, (ii) let the algorithm infer the priors using a hierarchical model and (iii) leverage covariate information to update these priors. I am not sure that my summary is a good one, but I think this difference with coloc could be made a bit clearer, perhaps expanding a bit in that first subsection of the results section.

Secondly, point (i) and (iii) are quite clear and seem well motivated, but I am not quite getting point (ii) about learning the priors from the data. What is the difference with a standard coloc that motivates that added feature? In other words, what is the information across many traits that motivates that extension? Or is the motivation coming from the "many variants" feature? It was not obvious to me what the shared information was? I think the penny eventually dropped for me later in the paper when query variants are classified into categories, which I suppose come with different priors, hence motivating the hierarchical prior idea. In any case, I think the rationale could be spelled out more clearly early on in the text to better guide the reader.

Thirdly, the authors mention MCMC on line 334 of the main text. There is hardly any reference to a MCMC algorithm in the paper. I found it in the supplemental methods where a Metropolis Hastings algorithm is described to manage the estimation of the hierarchical priors. Assuming I didn't miss it in the text, in which case I apologise, it would help to clarify when and why a MCMC type algorithm is required for the inference. Is that systematic for all definitions of these priors? What is the impact on computation time? And in general how fast does the method run?

As a more minor point, when the authors first talked about the genetic correlations, I initially understood it as leveraging the correlations between the query traits (such that, for example, if the phewas has multiple height GWAS, then the result of the analysis is consistent across these GWAS). What the authors leverage, I believe, is rather the genetic correlations between the primary trait and each of the query traits. I feel adding that "each" qualification on line 99 would help clarify what was done. And following on that, I wonder if the authors considered incorporating instead the full matrix of genetic correlations, and what the model would then look like? This is out of scope of this review but it may be worth a discussion item.

Reviewer #2 (Remarks to the Author):

This is a very interesting paper describing a new method for PheWAS, CoPheScan, that controls for LD confounding while doing PheWAS. This looks like it could be a very useful methodology for PheWAS, specifically reducing the downstream post-PheWAS analyses required to tease apart pleiotropy from LD contamination. Overall, I think this is an important advance and has a lot of potential.

While I am enthusiastic about the approach, the manuscript has a number of issues that result in a lack of clarity and general readability. There are some major points that leave me with questions about

the performance of the methodology and some minor concerns that should be considered to improve ease of reading/understanding the paper.

Major concerns:

- The discussion about configurations is not clear. After reading that section multiple times, I do not understand what is meant by " $Q + 1$ possible configurations", or "one configuration". Can the authors explain this concept more clearly so that readers can fully understand the implications of increasing/decreasing Q . Also, is there a difference between "one configuration" or "1 configuration"?
- Notation throughout the paper is not well defined. For example, on pages 2-3, what is " H_n ", " H_a ", " H_c ", " n ", " c ", " a ". These variables are never defined and thus I cannot follow the equations on page 3.
- On page 5, the authors say "The majority of these false positives were true H_a called as associated due to LD confounding." But isn't this what this method is trying to avoid? I thought CoPheScan was supposed to reduce the LD confounding? This made me think that I am not fully understanding the method.
- I think the Discussion would benefit from addressing the limitations of this approach. How much does the number of causal variants assumption impact the results? Will it cause inflation or deflation? False positives or negatives? What about diverse datasets with variable LD patterns? How will that impact? Do you need to stratify by global genetic similarity first?
- In the simulations, what 1000 Genomes populations were used? Was it all of them? Did they use the related people or only unrelated?
- Are the simulated datasets available for others to use and reproduce results?
- The simulations did only 1 or 2 causal variants. Then how do they generalize if there are 3-5 causal variants?

Minor concerns:

- On page 3, line 98, what is the r_g with an upside down question mark intended to represent?
- The way the authors represent the method changes throughout the paper from CoPheScan to cophescan. This is inconsistent and I think it would improve the quality of the manuscript to refer to the method the same way throughout.
- On page 8, what is r_g ? Also, should it be a subscript or just r_g ? Or are they different. In the same paragraph, the authors refer to "RA". I think that is meant to be rheumatoid arthritis, but since none of these are defined, I am guessing.
- Please double check genes are italicized throughout. I think they are most of the time, but sometimes it was unclear if you were referring to a gene or protein due to lack of italics.

REVIEWER COMMENTS

Reviewer #1 (Remarks to the Author):

I read with interest this paper by Manipur and colleagues that describes a method to query large scale GWAS type data for colocalisation with a variant associated with a trait of interest. This is a commonly asked question, often answered in very approximate ways, and this work provides an elegant and well motivated approach to answer the question. So there is absolutely no doubt in my mind that this paper should be published- this looks very useful and should receive a lot of attention/usage. My comments mostly relate to clarifications and presentation, which I think could help readers like myself.

1. My first request relates to better spelling out the methodological difference between the commonly used coloc methods (either as in Giambartolomei et al, or the SuSiE version) and this new algorithm. As I understand it, there are three modifications: (i) adapting coloc to a known causal variant situation, (ii) let the algorithm infer the priors using a hierarchical model and (iii) leverage covariate information to update these priors. I am not sure that my summary is a good one, but I think this difference with coloc could be made a bit clearer, perhaps expanding a bit in that first subsection of the results section.

Response:

Thank you for highlighting the need to provide more clarity in the methodological differences between coloc and CoPheScan. We have added the following in p3:

“Beyond the difference in the hypothesis space described above, CoPheScan differs from coloc in two further ways. First, because we have reduced the hypothesis space, we can examine many variants simultaneously, allowing us to learn the priors from the data in a hierarchical Bayesian manner with Markov Chain Monte Carlo (MCMC) sampling (Supplementary methods). In contrast, coloc assumes priors are fixed and known, which is a weakness because inference must rely on the investigators’ judgement on prior probabilities of colocalisation. Second, because we are using this hierarchical approach, we can exploit additional external information about the variants and/or the traits in the form of covariates which can be included when learning the priors. This allows the priors to vary depending on the query trait/query variant pairs being considered. Here, we include the genetic correlation (r_g) between the primary trait and each query trait tested (see Supplementary Methods).”

2. Secondly, point (i) and (iii) are quite clear and seem well motivated, but I am not quite getting point (ii) about learning the priors from the data. What is the difference with a standard coloc that motivates that added feature? In other words, what is the information across many traits that motivates that extension? Or is the motivation coming from the "many variants" feature? It was not obvious to me what the shared

information was? I think the penny eventually dropped for me later in the paper when query variants are classified into categories, which I suppose come with different priors, hence motivating the hierarchical prior idea. In any case, I think the rationale could be spelled out more clearly early on in the text to better guide the reader.

Response:

We hope that our additional clarifying paragraph in response to point 1 above also simultaneously addresses this point. The motivation is that it may be appropriate that priors vary between examples, and that where we have data that may inform that we should exploit it.

3. Thirdly, the authors mention MCMC on line 334 of the main text. There is hardly any reference to a MCMC algorithm in the paper. I found it in the supplemental methods where a Metropolis Hastings algorithm is described to manage the estimation of the hierarchical priors. Assuming I didn't miss it in the text, in which case I apologise, it would help to clarify when and why a MCMC type algorithm is required for the inference. Is that systematic for all definitions of these priors? What is the impact on computation time? And in general how fast does the method run?

Response:

Most Bayesian analysis requires MCMC sampling to sample from the posterior. Rather than discuss this, we have better signposted the supplemental Metropolis Hastings description, adding a subtitle "MCMC sampling from the posterior" to the Supplementary Methods, p5 .

We have added the following details regarding the computational time in the Supplementary Methods section on p6 along with a figure showing a linear increase in time with increase in either the number of variants/traits tested or the number of iterations:

Computational time

CoPheScan is implemented in R and C++ and is available on CRAN from <https://cran.r-project.org/package=cophescan>.

The computational time of the MCMC model is dependent on the size of the variant and trait sets considered. For 100 variants and 1000 traits, 100,000 iterations takes 25 minutes on a single Intel Ice Lake CPU of a high performance computing cluster. Timing scales linearly with the number of variant/trait pairs and the number of iterations as shown in Figure 1.

Figure 1: Computational time for the CoPheScan MCMC model in minutes (x axis) where the number of query variant - query trait (QV/QT) pairs (y axis) were varied and run for 100,000 iterations in (a) and the number of iterations (y axis) were varied and run with 100,000 QV/QT pairs in (b).

- As a more minor point, when the authors first talked about the genetic correlations, I initially understood it as leveraging the correlations between the query traits (such that, for example, if the phewas has multiple height GWAS, then the result of the analysis is consistent across these GWAS). What the authors leverage, I believe, is rather the genetic correlations between the primary trait and each of the query traits. I feel adding that " each" qualification on line 99 would help clarify what was done. And following on that, I wonder if the authors considered incorporating instead the full matrix of genetic correlations, and what the model would then look like? This is out of scope of this review but it may be worth a discussion item.

Response:

We agree that clarifying “each” on p3 is helpful, thank you.

“Here, we include the genetic correlation (r_g) between the primary trait and each query trait tested (see Supplementary Methods).”

It is an interesting idea to consider using the query trait correlation information in the model. As this contains no information about query variants, it cannot affect prior information in the current model (where priors are sampled independently), but perhaps could be used to penalise posterior samples where conclusions differ for highly correlated traits. We have added the following text to the Discussion:

“Our model can exploit covariate information that relates to query trait-query variant pairs, but would need to be extended to accommodate other information. For example, we might see modest evidence for causal

association of a medication trait with a given query variant, but intuitively trust the result is true because of stronger evidence at the same variant with the disease itself. *The difference in inference in such a case might be explained by the smaller numbers of individuals reporting use of a specific medication. We could consider exploiting genetic correlation between query traits by using a multivariate prior, with covariance linked to the genetic correlations. While this is beyond the scope of the current study, we hope our use of covariate-informed priors illustrates the potential for external information to be exploited when conducting PheWAS and other genetic studies.*"

Reviewer #2 (Remarks to the Author):

This is a very interesting paper describing a new method for PheWAS, CoPheScan, that controls for LD confounding while doing PheWAS. This looks like it could be a very useful methodology for PheWAS, specifically reducing the downstream post-PheWAS analyses required to tease apart pleiotropy from LD contamination. Overall, I think this is an important advance and has a lot of potential.

While I am enthusiastic about the approach, the manuscript has a number of issues that result in a lack of clarity and general readability. There are some major points that leave me with questions about the performance of the methodology and some minor concerns that should be considered to improve ease of reading/understanding the paper.

Major concerns:

1. The discussion about configurations is not clear. After reading that section multiple times, I do not understand what is meant by "Q + 1 possible configurations", or "one configuration". Can the authors explain this concept more clearly so that readers can fully understand the implications of increasing/decreasing Q. Also, is there a difference between "one configuration" or "1 configuration"?

Response:

To illustrate the meaning of the configurations we added Supplementary figure 1:

Supplementary Figure 1: The three CoPheScan hypotheses: H_n , H_a and H_c and their corresponding SNP configurations in a genomic region with Q SNPs (circles). The first SNP here is the query variant and the SNPs are shaded red when causal for the query trait.

We also clarified the definition of configuration in p2 of the manuscript:

“In a genomic region with Q SNPs, and under the initial assumption of a single causal variant (which we will relax later), there are $Q+1$ possible ways or “configurations” to describe where the single causal variant may lie, each corresponding to exactly one of three hypotheses (Supplementary Figure 1).”

"1 configuration" and "one configuration" are the same and have now been changed to "one" so as to be consistent throughout.

2. Notation throughout the paper is not well defined. For example, on pages 2-3, what is "Hn", "Ha", "Hc", "n", "c", "a". These variables are never defined and thus I cannot follow the equations on page 3.

Response:

H_n , H_a and H_c are the three hypotheses between which CoPheScan discriminates and are defined in p2 and 3. We made minor changes so the definitions of the hypotheses on pages 2 and 3 are clear making the subsequent equations easy to follow.

*“ H_n : No association of any variant with the query trait (one configuration)
 H_a : Causal association of a variant other than the query variant with the query trait ($Q-1$ configurations)
 H_c : Causal association of the query variant with the query trait (one configuration)”*

3. On page 5, the authors say " The majority of these false positives were true Ha called as associated due to LD confounding." But isn't this what this method is trying to avoid? I thought CoPheScan was supposed to reduce the LD cofounding? This made me think that I am not fully understanding the method.

Response:

Here we compare the CoPheScan method to conventional PheWAS approaches. This line refers to the traits simulated as true Ha being predicted as associated using traditional PheWAS. We have modified this line (now in p6, line 157) to clarify that this is a comparison to other approaches:

"The majority of the false positives obtained from these conventional approaches were true Ha but called as associated due to LD confounding."

4. I think the Discussion would benefit from addressing the limitations of this approach. How much does the number of causal variants assumption impact the results? Will it cause inflation or deflation? False positives or negatives? What about diverse datasets with variable LD patterns? How will that impact? Do you need to stratify by global genetic similarity first?

Response:

Thank you for the comment. We have now addressed these in the discussion in p19:

"Our CoPheScan approach has some specific limitations. The signals obtained from the multiple variant assumption rely on the available LD information. Zou et al demonstrated that the performance of SuSIE degrades when presented with out-of-sample LD matrices¹⁴. In cases where in-sample LD matrices are not accessible, it is recommended to utilise out-of-sample LD from large reference panels. Our experience with SuSIE is that as the number of causal variants increase, the performance (ability to detect all causal variants and/or ability to detect the correct causal variants) may decrease. In the case that a true causal variant signal is missed for our query variant, as occurred in around 35% of our simulations with two causal variants, CoPheScan concludes Ha - ie a false negative. SuSiE is likely to miss a higher fraction of secondary causal variants when the true number of causal variants increase, which would be expected to lead to greater false negatives for CoPheScan. Importantly, we do not see any increase in false positives when simulating two compared to one causal variant Analysis of rare events in large samples with standard methods can cause bias in regression summary statistics. Here, we used careful QC, thresholding on the number of events / MAF, but a better approach would be to use methods specifically developed to deal with this such as REGENIE to generate input to CoPheScan. The current form of CoPheScan only allows single-ancestry studies which will be addressed in future iterations and allow an increase in power to detect rare variants."

5. In the simulations, what 1000 Genomes populations were used? Was it all of them? Did they use the related people or only unrelated?

Response:

We used all EUR samples from 1000 Genomes. This has now been clarified on p20.

6. Are the simulated datasets available for others to use and reproduce results?

Response:

We have uploaded the simulated summary statistics and processed files on figshare (link is now in the data availability section).

The code to reproduce the datasets and results are available here:

<https://github.com/chr1swallace/cophescan-manuscript-sim-summary-data> and <https://github.com/ichcha-m/cophescan-paper>.

7. The simulations did only 1 or 2 causal variants. Then how do they generalize if there are 3-5 causal variants?

Response:

The behaviour of CoPheScan in the case of more causal variants will depend on the accuracy of SuSiE to accurately detect those causal variants. If SuSiE detects signals for 3-5 variants with the same efficiency as 2 variants, then the simulation results will hold. If SuSiE is less efficient as the number of true causal variants increases, then that will impact CoPheScan. As in our response to point 4, we believe the overwhelmingly most likely effect would be to increase false negatives, either because SuSiE does not detect, or incorrectly detects secondary signals in a multiple causal variant region. We believe our additional text on p19, replicated in response to point 4, covers this case.

Minor concerns:

1. On page 3, line 98, what is the r_g with an upside down question mark intended to represent?

Response:

There was an error caused during the conversion to the pdf format and is meant to be r_g without the question mark. This formatting has been corrected.

2. The way the authors represent the method changes throughout the paper from CoPheScan to cophescan. This is inconsistent and I think it would improve the quality of the manuscript to refer to the method the same way throughout.

Response:

Corrected to CoPheScan throughout the paper, thank you.

3. On page 8, what is r_g ? Also, should it be a subscript or just rg ? Or are they different. In the same paragraph, the authors refer to "RA". I think that is meant to be rheumatoid arthritis, but since none of these are defined, I am guessing.

Response:

r_g is the genetic correlation between the primary trait and the query trait and is defined in p3, line 107 and p22, line 586 in the Methods section.

r_g should be used with a subscript, thank you for pointing this out. This has been corrected in a few places where this was not the case.

RA is defined as Rheumatoid arthritis in line 206.

4. Please double check genes are italicized throughout. I think they are most of the time, but sometimes it was unclear if you were referring to a gene or protein due to lack of italics.

Response:

This has been checked and corrected.

REVIEWERS' COMMENTS

Reviewer #1 (Remarks to the Author):

I appreciate the authors' reply to my comments. I have no further queries and I look forward to seeing the paper online.

Reviewer #2 (Remarks to the Author):

The authors did a very nice job responding to all of the comments from both reviewers. This is an excellent paper.